# Effects of coenzyme Q10 and N-acetylcysteine on experimental poisoning by paracetamol in Wistar rats

Rayanne Henrique Santana da Silva[1], Mariana de Moura[2], Larissa de Paula[3], Kelly Carolina Arantes[1], Marina da Silva[4], Jaqueline de Amorim[4], Marina Pacheco Miguel[3], Danieli Brolo Martins[4], Daniela de Melo e Silva[2], Marília Martins Melo[5], Ana Flávia Machado Botelho [1]*

1 Veterinary Toxicology Laboratory, Veterinary Hospital, Veterinary and Animal Science School, Federal University of Goiás, Goiânia, Goiás, Brazil, 2 Department of Genetics, Laboratory of Mutagenesis, Federal University of Goiás, Goiânia, Goiás, Brazil, 3 Pathology Sector, Instituto de Patologia Tropical e Saúde Pública, Federal University of Goiás, Goiânia, Goiás, Brazil, 4 Veterinary Clinical Pathology Laboratory, Veterinary Hospital, Veterinary and Animal Science School, Federal University of Goiás, Goiânia, Goiás, Brazil, 5 Veterinary Toxicology Laboratory, Veterinary Hospital, Veterinary and Animal Science School, Federal University of Minas Gerais, Belo Horizonte, Minas Gerais, Brazil

* anafmb@ufg.br

**Data Availability Statement:** All relevant data are within the paper and its Supporting Information files.

## Abstract

Paracetamol (PAR) is a drug widely used in human and veterinary medicine as an analgesic and antipyretic, often involved in cases of intoxication. The most common clinical signs result from damage to red blood cells and hepatocytes, and this intoxication is considered a model for the induction of acute liver failure. In the present study, the hepatoprotective effects of coenzyme Q10 (CoQ10) and N-acetylcysteine (NAC) against experimental paracetamol (PAR) poisoning were analysed. Thirty-five adult Wistar rats (*Rattus novergicus albinus*) were randomly assigned to five groups, and thirty-one of these survived the treatments. Negative control group (CON-) received 1mL of 0.9% NaCl orally (PO). Other groups received 1.2g/kg of PAR (PO). Positive control group (CON+) received only PAR. NAC group received 800 mg/kg intraperitoneally (IP) of NAC 1h after the administration of PAR and at 12 h received 1mL of 0.9% NaCl, IP. The fourth group (CoQ10) received 1h and 12 h after intoxication, CoQ10 (10mg/kg IP). And the fifth group (NAC+CoQ10) received NAC (800mg/kg, IP) and CoQ10 (10mg/kg, IP). After 12 hours, the rats were euthanized and necropsied to collect liver and kidney tissues for histopathological evaluation and electronic microscopy. A single dose of PAR caused severe acute hepatitis. NAC couldn't reverse the liver and kidney damages. The group that received CoQ10 and NAC had moderate liver damage, while the group that received only CoQ10 had lower values of liver enzymes and mild liver and kidney damage. Animals that received treatment with CoQ10 or NAC+CoQ10 presented normal hepatocyte mitochondria and nuclei. Although CoQ10 couldn't reverse PAR organ damage, results indicate promising hepatoprotection in Wistar rats.

**Funding:** We wish to acknowledge "Coordenação de Aperfeiçoamento de Pessoal de Nível Superior"—Finance Code 001" for the fellowship to Rayanne Henrique Santana da Silva. Also, we acknowledge the financial support for publication from FAPEG (Research Support Foundation of the State of Goiás), grant 06/2018.

**Competing interests:** The authors have declared that no competing interests exist.

## Introduction

Paracetamol (PAR) is an analgesic drug with antipyretic activity, widely marketed and used in human and veterinary medicine [1–3]. Inadequate doses and/or used in contraindicated species promote acute liver failure and death. Its mechanism of action involves liver metabolism, as high levels of N-acetyl-p-benzoquinone imine (NAPQI) are formed in the P450 cytochrome, resulting in glutathione depletion and oxidative damage [4, 5]. To reverse or reduce oxidative damage, two drugs have been studied for their therapeutic potential, N-acetylcysteine (NAC) and Coenzyme Q10 (CoQ10).

NAC acts as a replenishing donor of glutathione and sulphur, for the antioxidant cycle. For a long time, NAC has been used not only in cases of paracetamol intoxication, as an antioxidant, but also as an expectorant agent and in the treatment of keratitis and dry eye disease. New research refers to antimicrobial action against bacterial keratitis, interruption of biofilm formation and corneal healing [6], in addition to heart disorders caused by the use of doxorubicin [7]. However, although NAC is the drug of choice in PAR poisoning, coenzyme Q10 has successfully treated poisoning and disease [8, 9].

CoQ10 is a non-enzymatic, liposoluble antioxidant, consisting of a benzoquinone ring and a species-specific isoprenoid chain, being synthesized endogenously by aerobic organisms [10, 11]. CoQ10 participates in the mitochondrial respiratory chain, in oxidative phosphorylation, as a carrier of electrons and protons [12]. Due to its antioxidant action against the excessive production of free radicals, recent studies have shown promising therapeutic results. This action occurs by eliminating and preventing the initiation and propagation of lipid peroxidation. In addition, it can also interrupt the deleterious action of cytokines such as tumor necrosis factor (TNF-α) and interleukin 6 (IL-6) in inflammatory processes [13]. Coq10 has been studied regarding its action on cardiac tissue exposed to oxidative stress, preventing further damage to the myocardium and contractile function of the heart [7].

In addition, it is possible to associate its antioxidant effects in different systems of the organism, such as the digestive and neurological systems [14]. Considering the antioxidant and anti-inflammatory effect of CoQ10, the present study aimed to evaluate the therapeutic potential of this cofactor and its association with NAC in the treatment of acute liver failure induced experimentally by PAR in rats.

## Materials and methods

### Experimental design

All experimental procedures were approved by the Ethics Committee for the Use of Animals (CEUA) of the Federal University of Goiás (062/19). Thirty-five adult male Wistar rats (*Rattus novergicus albinus*) were used, weighing on average 200-300g, distributed in five groups, and thirty-one of these survived the treatments. Animals were acquired at the Federal University of Goiás Bioterium.

The first group, called negative control (CON-) (n = 6), received orally (PO) a single dose, 1mL of 0.9% NaCl. The other groups received 1.2g/kg of PAR, PO (Tylenol® drops, 200 mg/mL, Lot: AM4000). The second group, positive control (CON+) (n = 7), received only PAR as above. The third group (NAC) (n = 5) received, one hour after PAR, as treatment, an intraperitoneal (IP) injection of 800mg/kg of N-acetylcysteine (Fluimucil®, ampoule, 100 mg/mL). Followed by 0.9% NaCl, IP after 12 h. The fourth group (CoQ10) (n = 6) received in hours one and 12 after intoxication: CoQ10, 10mg/kg, IP. And the fifth group (NAC+CoQ10) (n = 7) received the association of: N-acetylcysteine (800mg/kg IP) and CoQ10 (10mg/kg IP) one hour after PAR administration, followed by NaCl 0.9% IP and 10mg/kg CoQ10 IP, after 12h.

After 12 hours of treatment, an accumulation of 24 hours of poisoning, all animals were euthanized by deepening anaesthetic with 5% isoflurane (Isoflurane®). After that, blood was collected followed by necropsy with organ sampling for histopathological examination, electron microscopy, and comet assay.

### Clinical evaluation

Behavioural changes were evaluated every hour by a single evaluator, from poisoning to the end of the experiment. General activity, physical and mental status, behaviour, and possible deaths were observed.

### Haematological evaluation

Blood was collected puncturing the abdominal aorta, during the anaesthetic procedure, and distributed in a 0.5 mL tube containing 10% EDTA anticoagulant and in a 2 mL tube with clot activator gel, to obtain serum.

Samples were immediately processed at the Veterinary Clinical Laboratory of UFG. The CBC (cell blood count) count was performed by an automatic analyser (Celltac α MEC 6550, Nihon Kohden®, Japan). This provided parameters of white blood cells (WBC); red blood cell (RBC); haemoglobin (HGB); mean corpuscular volume (MCV); mean corpuscular haemoglobin (MCH); mean corpuscular haemoglobin concentration (MCHC); red cell distribution width standard deviation (RDW-SD); coefficient of variation of erythrocyte amplitude (RDW-CV); platelet count (PLT); platelet distribution width (PDW); and mean platelet volume (MPV). The haematocrit and total protein (TP) were manually analysed, respectively by the microhematocrit method and the refractometer. Blood smears and panoptic fast staining were performed to assess cell morphology and leukocyte differential count. The same evaluator performed both.

### Serum biochemistry

For the biochemical evaluations, blood collected in tubes without anticoagulant was centrifuged for 5 minutes in a rotation of 3600 rpm, to obtain the serum. The biochemical profile was measured through alanine aminotransferase (ALT), aspartate aminotransferase (AST), alkaline phosphatase (ALP), gamma-glutamyl transferase (GGT), total-value bilirubin (TBIL), direct bilirubin (DBill) and indirect bilirubin (IBIL), total serum protein (TSP), albumin (ALB), globulin (GLB), serum urea and creatinine by the automatic biochemistry device (CM 250, Wiener®, Argentina), using commercial kits (Biotécnica® and Doles®).

### Histopathology

After euthanasia, necropsy and macroscopic evaluations of all animals were performed. Kidney and liver fragments were fixed in 4% paraformaldehyde. These fragments underwent routine histological processing for paraffin embedding, following steps of dehydration, diaphanization, and paraffin infiltration. After making the paraffin blocks containing tissue fragments, 4 μm-thick histological sections were performed. Later, the histological slides were stained by the haematoxylin-eosin (HE) stain.

Cuts and preparation were performed at the Institute of Tropical Pathology and Public Health at UFG. For histological evaluation, liver scores ranged from zero to four. As for the kidneys, scores went from zero to three, as shown in Table 1.

**Table 1. Scores for hepatic and kidney injuries from Wistar rats subjected to experimental poisoning by acetaminophen and treated with Coenzyme Q10 (CoQ10), N-acetylcysteine (NAC) and CoQ10 associated with NAC.**

| Microscopic finding in liver tissue | Score |
|---|---|
| Hepatocyte atrophy or absence of changes in more than 70% of the slice | 0 |
| Presence of hydropic degeneration/Individual necrosis (apoptosis)/Absence of nuclei in more than 70% of the cut | 1 |
| Zonal hepatitis in over 70% of the cut | 2 |
| Coagulative or lytic zonal necrosis in more than 70% of the cut | 3 |
| Confluent lytic bridge necrosis (sub massive or massive) in more than 70% of the cut | 4 |
| **Microscopic finding in kidney tissue** | **Score** |
| No changes/Hyperaemia/Slight intratubular protein/Slight vesiculation/Glomerular space enlargement/Slight edema | 0 |
| Moderate perivascular edema/moderate blistering/moderate intratubular protein/moderate glomerular space enlargement | 1 |
| Mild vascular thinning / Mild degeneration / Mild glomerular protein / Moderate tubular vesicle / Mild dilatation / Mild debris / Moderate intratubular protein | 2 |
| Moderate and severe vascular thinning/Moderate and severe degeneration/Moderate and severe glomerular protein/Severe tubular vesicle | 3 |

## Electron microscopy

Samples of about 1 mm thick from the liver and kidneys were sent for evaluation by electron microscopy at the Center for Acquisition and Processing of Images at UFMG (CAPI-ICB/UFMG) and the Center for Microscopy at UFMG (CM-UFMG), fixed in 2% glutaraldehyde and kept in 1.5M phosphate buffer under freezing until use.

## Comet assay

The comet assay was performed from whole blood samples obtained after euthanasia. The tubes of cell suspensions were centrifuged at 100 rpm for 10 min. After, 15 µL of this cell suspension were removed and soaked in 120 µL of Low Melting Agarose (0.5%), remaining in a water bath at 37˚C. This set was put in a slide previously prepared with a precoat of Normal Melting Agarose (1.5%) and covered with a coverslip. The slides were placed in a refrigerator at 4˚C for ten minutes to solidify and the coverslips removed. The slides were immersed in lysis solution (1 mL Triton X-100, 10 mL DMSO, and 89 mL stock lysis solution, pH 10.0) for 24 hours.

After this time, the slides were removed from the lysis solution. Neutralisation was performed with a buffer solution (Tris-HCl, 0.4 M, pH 7.5) three times for 5 min. The slides were captured by fluorescence microscopy (ZEISS, 516–560 nm filter, and 590 nm filter barrier) at 100x magnification. For the evaluation of DNA damage, two main parameters were calculated: percentage of DNA in the tail (%DNA) and Olive Tail Moment (Olive Tail Moment—MCO) with TriTek Comet ScoreTM software.

## Statistical analysis

The experimental design used was completely randomised. The data presented as mean and standard deviation. Kolmogorov-Smirnov test was performed to define normality. ANOVA and Tukey's post-test evaluated the parametric variables. In the case of non-parametric variables, Kruskal-Wallis and post-Dunn's multiple comparisons test were used. Spearman's correlation test was used to assess the relationship between serum biomarker levels and histological lesion scores. In all analyses, a significant difference was considered when $p < 0.05$. Data were evaluated using GraphPad Prism 8.

## Results

### Clinical evaluation

It was observed that all animals in the groups that received PAR presented apathy. Although The CON+ group did not receive any treatment, piloerection, ataxia after stimulation, and diarrhoea were observed. No deaths occurred in this group.

### Haematological and serum biochemistry evaluation

The administration of PAR induced an increase in haematocrit, erythrocytes and haemoglobin, in the CON+ group and evidenced thrombocytopenia in the group of animals intoxicated with PAR and treated with NAC (Fig 1). Liver function was assessed through total serum protein (TP), its fractions (albumin and globulin), and total, direct and indirect bilirubin. It was observed that a decrease in proteins in treated groups compared to the positive control, partially associated with decreasing albumin levels (Fig 1). No significant difference was observed in globulins, ALT or bilirubin (S1 Fig).

To evaluate the liver function ALT and AST were measured. No significant differences were observed between groups regarding ALT (Supporting material). However, the AST assessment revealed a significant increase in the CON+ group (7668 ±10980 U/L) concerning the CON- (218.2 ±21.99 U/L) and NAC (5897 ±5203 U/L) groups (Fig 2). Creatinine and urea were measured to assess the renal profile. Significant increase in creatinine was observed in the NAC group compared to the NAC+CoQ10 group (Fig 2).

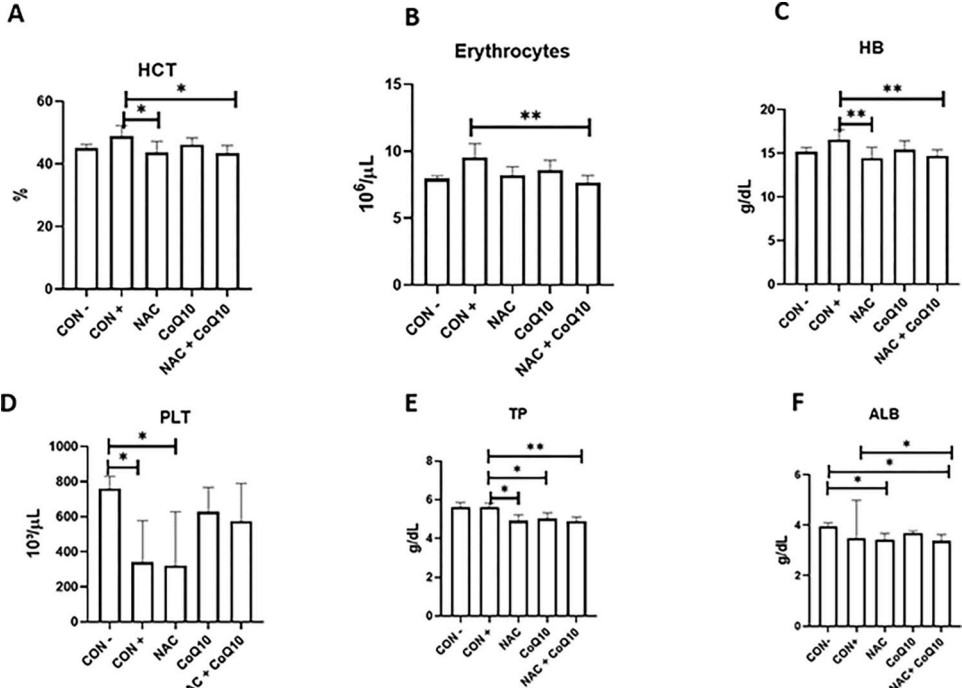

**Fig 1.** (A) Mean and standard deviation of haematocrit (HCT), (B) erythrocytes and (C) haemoglobin, (D) platelets (PLT), (E) total serum protein (TP), (F) albumin (ALB) of rats experimentally poisoned with paracetamol and treated with coenzyme Q10 and N-acetylcysteine. CON-: negative control; CON+ positive control; NAC group treated with N-acetylcysteine; CoQ10 group treated with coenzyme Q10; NAC+CoQ10: group treated with the association of NAC and CoQ10. Analyses performed with Tukey or Kruskal-Wallis post-test (*p<0.05; **p<0.01).

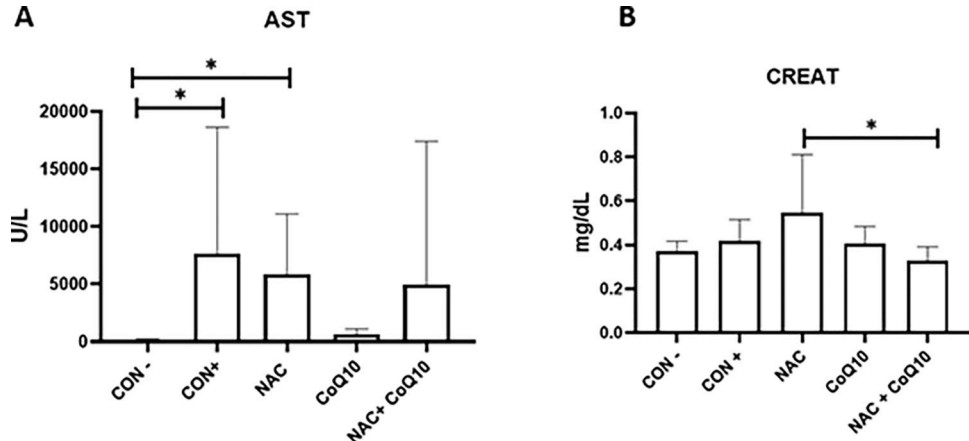

**Fig 2.** (A) Mean and standard deviation values of aspartate aminotransferase (AST) and (B) creatinine of rats experimentally poisoned with paracetamol and treated with coenzyme Q10 and N-acetylcysteine. CON-: negative control; CON+ positive control; NAC group treated with N-acetylcysteine; CoQ10 group treated with coenzyme Q10; NAC+CoQ10: group treated with the association of NAC and CoQ10. Analyses performed with Tukey or Kruskal-Wallis post-test (*p<0.05; **p<0.01).

## Comet assay

The evaluation of DNA damage was calculated by the parameters' percentage of DNA in the tail (%DNA) and Olive Tail Moment (Olive Tail Moment—MCO). The results obtained by the analysis show the mean ± SD (standard deviation) of the groups, where there was no statistical difference between them.

## Macroscopic evaluation

Regarding the macroscopic evaluation, marked multifocal congestion and evidence of a diffuse lobular pattern was seen in the CON+ group. In the CoQ10 group, slightly bulging edges, evidence of a moderate multifocal lobular pattern, and mild multifocal congestion are observed (Fig 3). The other groups showed no changes.

## Microscopic evaluation

Microscopically, the liver of the CON- group appeared normal (Fig 4). In the CON+ group, lesions characteristic of PAR intoxication were observed with moderate hyperaemia, necrosis, and bridging lytic haemorrhage in the centrilobular region marked in more than 70% of the evaluated cut in most of the evaluated animals (Fig 4).

The group that received 800mg/kg IP of NAC as treatment presented marked hyperaemia, hepatocyte necrosis, and marked haemorrhage in the centrilobular region. Animals that received only CoQ10 (10mg/kg, IP) showed mild hyperaemia and an absence of hepatocyte nuclei in the centrilobular region. The group treated with NAC + CoQ10 had moderate hyperaemia, mild atrophy of hepatocytes, and absent hepatocyte nuclei, predominantly in the centrilobular region (Fig 5).

In a few animals, there were areas of multifocal nephrosis. In all groups intoxicated with PAR, renal lesions showed in less than 50% of the tissue evaluated. On the other hand, the animals in the groups treated with NAC showed moderate to a marked increase in glomerular space, thinning of the endothelial basement membrane, dilation of glomerular capillaries, glomerular degeneration, vesiculation of tubular cells, tubular dilation, and presence of cellular debris in the intratubular space. The CoQ10 group showed only discrete intratubular

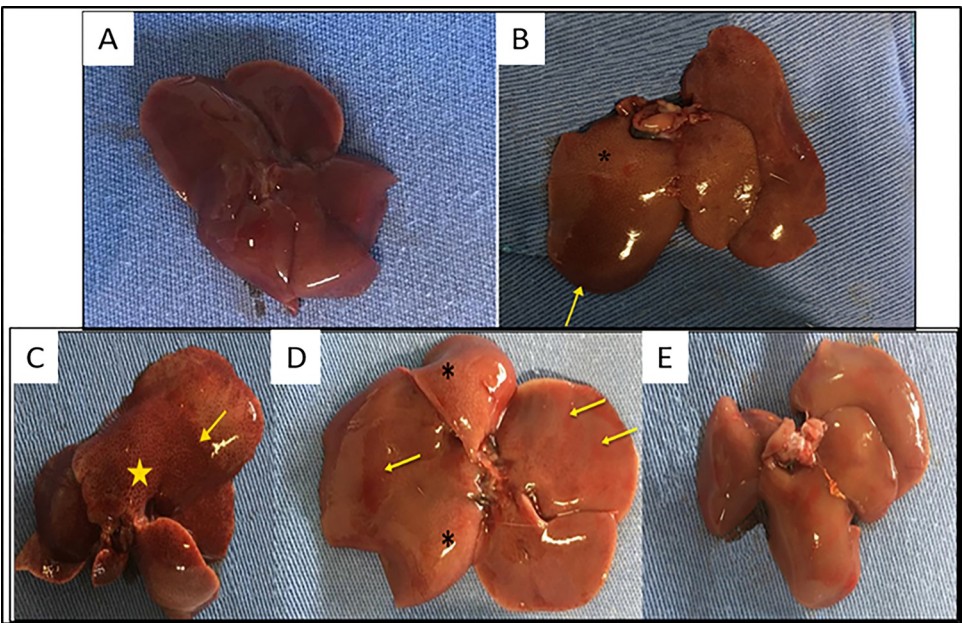

**Fig 3. Liver of Wistar rats.** (A) Photograph of the liver of a rat in the CON- group that received a single dose of 1mL of 0.9% NaCl PO, with normal pattern. (B) Photograph of rat liver from the CON+ group, only poisoned with paracetamol (PAR) 1.2 g/kg, PO. Multifocal and marked congestion (yellow arrow) and evidence of a marked and diffuse lobular pattern (asterisks) are visualized. (C) Photograph of the liver of a rat in the NAC group, intoxicated with PAR and treated with 800 mg/kg of NAC, with marked evidence of a diffuse lobular pattern (star) and extensive multifocal congestion (yellow arrow). (D) Photograph of rat liver from the CoQ10 group that received 10 mg/kg CoQ10, IP, showing slightly bulging edges, evidence of a moderate multifocal lobular pattern on the image (asterisks) and mild multifocal congestion (yellow arrows). (E) Photograph of the liver of the mouse from the NAC+CoQ10 group, treated with 800 mg/kg of NAC and 10 mg/kg of CoQ10, IP, without visible macroscopic changes.

degeneration. The NAC + CoQ10 group showed a discrete to moderate increase in glomerular space, glomerular degeneration, and intratubular protein (S2 Fig).

Among the groups evaluated, the CON+ and the NAC had the highest scores of injuries to liver tissue, higher than the CON- group. The CON+, NAC+CoQ10 and NAC groups showed higher injury scores in the assessment of renal tissue.

Spearman's correlation verified the relationship between liver and kidney injury scores and serum biochemistry, AST, urea, and creatinine. There was a weak correlation between creatinine scores and kidney damage (Table 2). Regarding urea, there was a very weak correlation for the CON- group, weak for NAC+CoQ10 and moderate for CON+, NAC and CoQ10 (Table 2).

And for the AST variable, a moderate correlation was observed for the CON- and NAC groups, a strong correlation for the CON+ and CoQ10 groups, and a very strong correlation between the same variable and the NAC+CoQ10 group (Table 2). Considering all values analysed, the correlation between AST values and liver histological score was extremely high (0.914).

## Ultrastructure evaluation

The ultrastructural analysis of kidney and liver was performed. This further evaluates the hepatotoxicity and nephrotoxicity of PAR. The animals in the CON- group showed unaltered liver tissue, with a vast presence of intact mitochondria, rough endoplasmic reticulum, and hepatocytes with normal nuclei, in addition to the fat droplets on the slide. In contrast, the positive

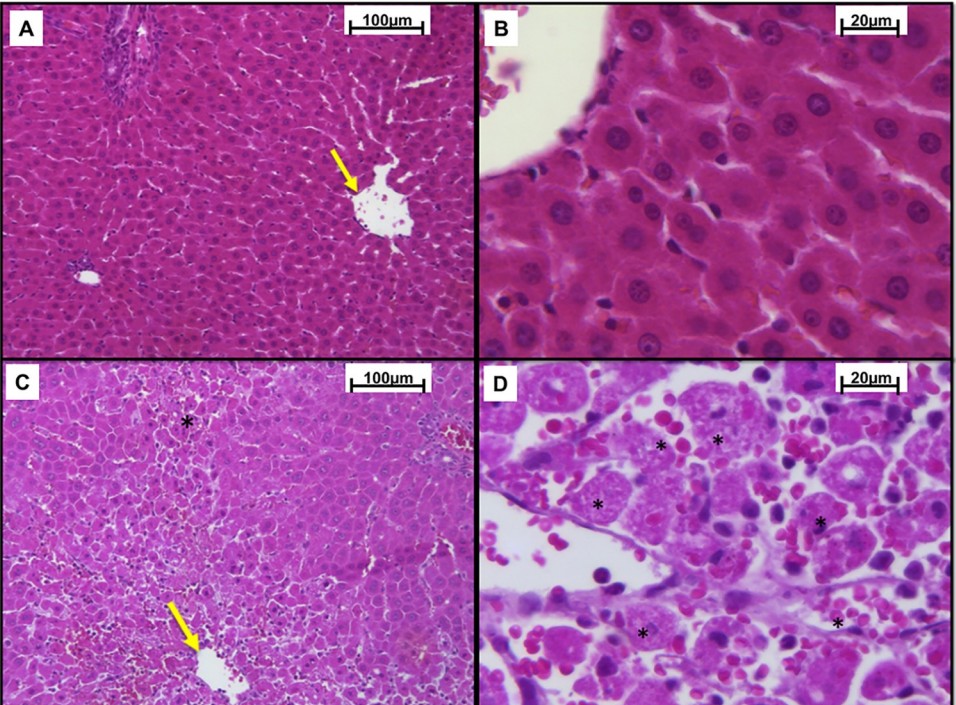

**Fig 4. Optical photomicrograph of liver samples from Wistar rats.** (A, B) Negative control group (CON-) presents mild hyperaemia in the centrilobular vein region (yellow arrow). (C and D) Positive control group (CON+), intoxicated with paracetamol (PAR) 1.2 g/kg, orally, presents moderate hyperaemia (yellow arrow) and necrosis and lytic haemorrhage in bridging in the centrilobular region marked (asterisk) (HE, Bar = 100 μm); (B and D) (HE, Bar = 20 μm).

control group, which received 1.2g/kg of PAR orally as the only treatment, had many collagen fibres, degenerated hepatocytes, red blood cells, and a lot of fibrosis throughout the tissue. Association of CoQ10 and NAC and CoQ10 alone had intact collagen, preserved mitochondria, and normal hepatocyte nuclei (Fig 6). And the renal tissue, in turn, presented intact tubules and glomeruli in all groups (S3 Fig).

## Discussion

In the present study, the protective effects of coenzyme-Q10 were determined against PAR toxicity in Wistar. PAR is one of the most used analgesics worldwide. It frequently causes poisoning in humans and animals [15, 16]. PAR intoxication is also a known model for acute liver damage [17]. Several drugs have been used to prevent and treat the toxic effects of PAR. Q10 is considered one of the most prominent [18]. However, this is the first study that evaluates the treatment of CoQ10 in comparison and association with NAC.

The PAR dose administered was based on previous studies that advocated between 1g/kg and 1.2g/kg [19–23]. During a pilot project, we concluded that 1.2g/kg caused the most significant clinical signs of poisoning, such as prostration, piloerection, ataxia after stimulation, and diarrhoea. Similar clinical signs are reported in intoxicated animals [24, 25] Thus, the dosage employed was sufficient to cause clinical intoxication, without causing death.

Most clinical manifestations are associated with PAR effects on the liver tissue. After oral intake, PAR is absorbed in the gastrointestinal tract, and transported to the liver, where it undergoes biotransformation. PAR can follow three main pathways: conjugation with

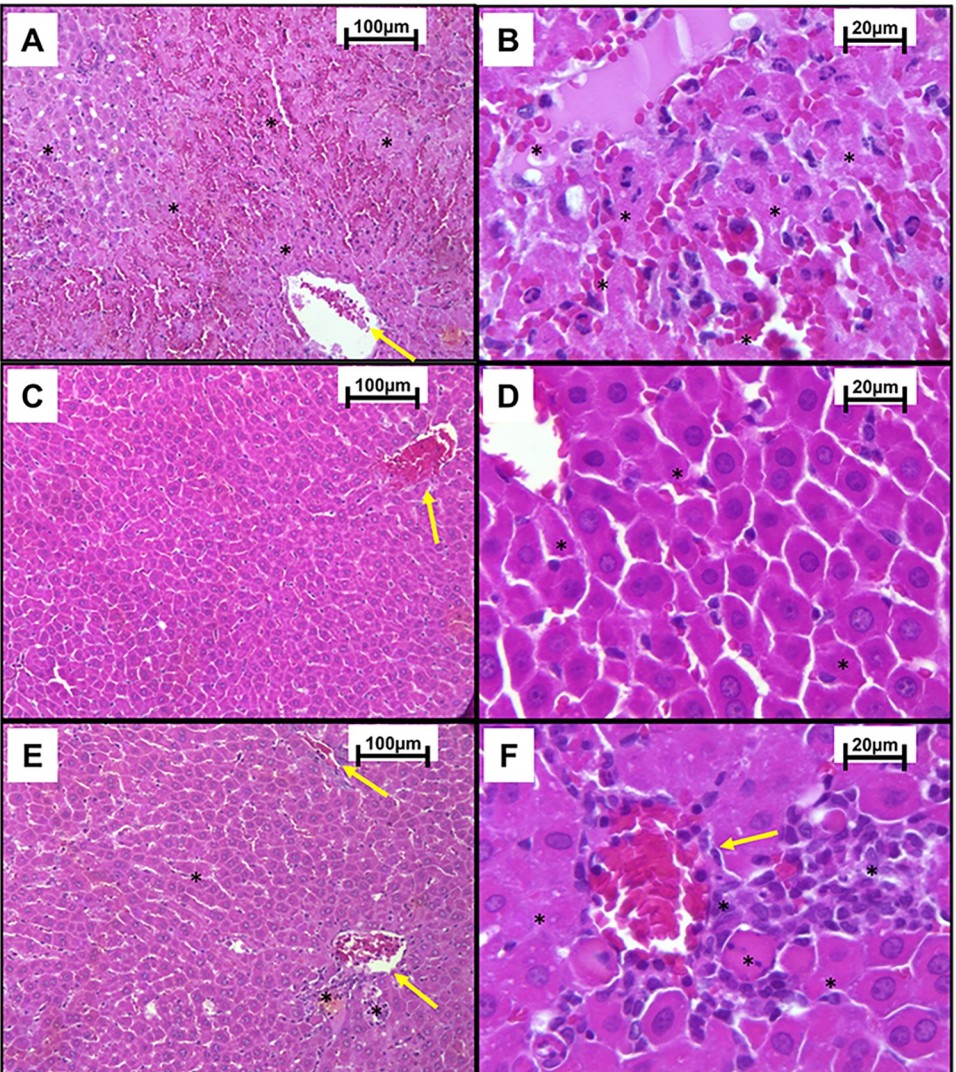

**Fig 5. Photomicrograph of liver tissue from Wistar rats intoxicated with 1.2 g/kg, PO, of paracetamol (PAR).** (A, B) The group that received 800mg/kg IP of N-acetylcysteine (NAC) presented marked hyperaemia (yellow arrows), necrosis of hepatocytes and haemorrhage in the centrilobular region marked in bridge (asterisks). (C, D) Group that received only 10mg/kg IP of Coenzyme Q10 (CoQ10) presented mild hyperaemia (yellow arrows) and absent hepatocyte nucleus (asterisk). (E, F) NAC + CoQ10 group, which were treated with 800 mg/kg of NAC and 10 mg/kg of CoQ10, IP and presented moderate hyperaemia (yellow arrow), mild atrophy of hepatocytes and absence of hepatocyte nucleus (asterisk). (A, C and E) (HE, Bar = 100 μm); (B, D and F) (HE, Bar = 20 μm).

**Table 2. Values of the Spearman's correlation coefficient between the variables creatinine, urea and AST and renal and hepatic injury scores in rats intoxicated with paracetamol and treated with coenzyme Q10 and N-acetylcysteine.**

| Variables | Spearman's correlation | | | | |
|---|---|---|---|---|---|
| | **CON-** | **CON+** | **NAC** | **Q10** | **NAC+Q10** |
| **Creatinine** | 0.00 | 0.03 | 0.25 | -0.16 | 0.14 |
| **Urea** | 0.00 | -0.43 | 0.44 | -0.42 | 0.31 |
| **AST** | 0.48 | 0.77 | 0.58 | 0.87 | 0.95 |

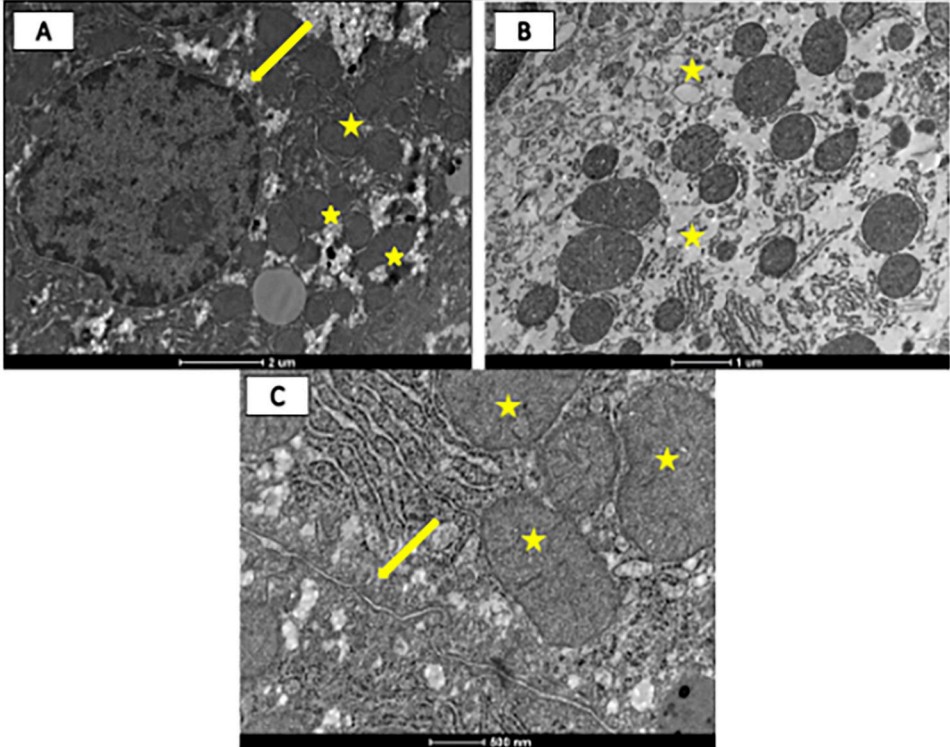

**Fig 6. Electron micrographs of longitudinal sections of liver tissue from rats experimentally intoxicated by paracetamol (PAR) and treated with coenzyme Q10 and N-acetylcysteine.** (A) CON- sample: emphasis on intact mitochondria (yellow star), normal hepatocyte nuclei (yellow arrow) and fat droplets; (B) CON+: fibrous tissue, degenerated mitochondria (yellow star); (C) NAC + CoQ10 group that were treated associated with NAC and CoQ10: emphasis on collagen fibers (yellow arrow) and intact mitochondria (yellow star).

glucuronic acid, conjugation with sulphates, and oxidation by cytochrome P450 [4, 25]. However, excessive doses of PAR overload the biotransformation system, increasing the oxidation process, and generating a toxic metabolite known as NAPQI. This metabolite saturates the hepatoprotective system of GSH, accumulates in the body, and reacts with unsaturated proteins and lipids, causing hepatocyte necrosis [15, 25].

PAR liver damage was confirmed by severe AST increase and histological and ultrastructural lesions, such as significant degeneration and necrosis, as previously reported [10]. The comet assay was used to identify DNA lesions caused by PAR. This assay is a sensitive test capable of evaluating the genotoxic potential of eukaryotic cells. Although PAR overdose produces ROS and changes DNA, and lipid peroxidation [10, 26], in the present study, no differences were observed between the groups. It is possible to conclude that the single dose of PAR used, as the treatments applied were not enough to change the cellular DNA.

An increase in haematological values was attributed to relative polycythaemia, as animals presented diarrhoea and prostration [27, 28]. Reduced platelet count could also be associated with PAR hepatocyte necrosis and haemorrhage [27, 29]. No significant alterations were observed in GGT, alkaline phosphatase (ALP), and ALT. This was expected since ductal damage is not a common finding in PAR poisoning [22, 30] and individual responses are associated with genetic factors [31, 32].

NAC treatment alone did not reverse the main alterations. Animals presented high levels of AST and creatinine, with significant liver necrosis and haemorrhage, and the highest level of

hepatic lesion according to our proposed score system. Despite NAC being a drug recommended in cases of PAR poisoning [9, 33, 34], It was observed that its administration alone had no hepatoprotective effect. NAC is a prodrug, precursor of glutathione and, therefore, has antioxidant activities. However, its mechanism of action is correlated with the early stage of liver injury, and with individual patient responses, that is, if NAC is administered late, after established liver damage, it will not have the same effectiveness [35]. Its effectiveness correlates with the time to treatment and individual glutathione stores [9, 36]. CoQ10 demonstrated hepatoprotection in isolation and association with NAC, minimising the toxic effects of PAR on hepatocytes.

CoQ10 alone induced lower levels of AST and liver damage, without incremental effects, when associated with NAC. AST is a mitochondrial and cytoplasmic enzyme, present in the deamination reaction of aspartate to oxaloacetate, and exhibits significant concentrations in the liver [30]. The elevation of this parameter is related to the damage to hepatocytes. AST presented the highest correlation to liver tissue damage and should be considered a biomarker for PAR poisoning.

Both NAC + CoQ10 and CoQ10 groups only caused discrete lesions in hepatocytes, without evolution to necrosis. Also, it is not clear why CoQ10 + NAC would be worse than CoQ10 alone for AST.

We believe this effect is due to CoQ10's ability to act as an antioxidant against NAPQI damage. Previous studies associated PAR poisoning with reduction of hepatic CoQ10 levels and that its supplementation through treatment can reduce ROS levels and ameliorate liver injury [18]. CoQ10 can increase nuclear factor erythroid 2-related factor 2 (Nrf2) expression and activity, which enables antioxidant defence against PAR [18]. CoQ10 action is also associated with its physiological role in the mitochondrial respiratory chain, as the supplementation enhances mitochondrial function and beneficial mitophagy [18]. CoQ10 can suppress NO production, prevent stress-induced nitrogen release, and reduce proinflammatory cytokines [10]. Thus, it is possible to hypothesise that CoQ10 reduces the deleterious oxidative effects of NAPQ1, a toxic metabolite of PAR, promoting beneficial antioxidant effects [11, 37].

Contrary to our initial beliefs, NAC and CoQ10 association did not improve the PAR intoxication treatment. As they act in different antioxidant pathways, our results suggest their effect is not synergic.

It is important to highlight that the study also demonstrates that the proposed liver histological scores can assess acute hepatitis associated with PAR, with a high correlation with AST, a biochemical parameter that showed better discrimination between groups. Likewise, Spearman's correlation demonstrated a strong relationship between high levels of AST and liver injury scores in the CON+ group. Despite the low specificity of AST, it is in the hepatocyte's mitochondria, and its increase is related to the severity of liver damage [38]. The strong correlation between the two parameters mentioned above confirms the use of AST measurement as a prognostic factor in animals intoxicated by PAR.

## Conclusions

The present research reaffirmed the beneficial use of CoQ10 against PAR toxicity, demonstrating expressive results in haematological, biochemical, histological, and ultrastructural parameters, proving that the coenzyme is promising. Treatment with CoQ10 minimised liver damage, possibly by reducing oxidative stress and lipid peroxidation induced by PAR, with no added benefits to its association with NAC. Our results suggest that treatment with this coenzyme exerts a hepatoprotective effect against PAR poisoning.

## Supporting information

**S1 Fig.** (A) Mean and standard deviation of globulin (GLOB), (B) ALT, (C) total bilirubin, (D) indirect bilirubin, (E) direct bilirubin, of rats experimentally poisoned with paracetamol and treated with coenzyme Q10 and N-acetylcysteine. CON-: negative control; CON+ positive control; NAC group treated with N-acetylcysteine; CoQ10 group treated with coenzyme Q10; NAC +CoQ10: group treated with the association of NAC and CoQ10. Analyses performed with Tukey or Kruskal-Wallis post-test (*$p < 0.05$; **$p < 0.01$). No significant difference was observed regarding the globulins, ALT and total bilirubin, direct bilirubin, indirect bilirubin parameters. (TIF)

**S2 Fig. Photomicrograph of renal tissue from Wistar rats intoxicated with 1.2 g/kg, PO, of paracetamol (PAR).** Negative control group (CON-) with normal aspect. (C) Positive control group (CON+), intoxicated with paracetamol (PAR) 1.2 g/kg, PO, presents a slight increase in the glomerular space (yellow arrow), tubular dilation (asterisk) and presence of intratubular protein; (D) necrosis and detachment of tubular cells (asterisks). (A and C) (HE, Bar = 20 μm). (PNG)

**S3 Fig. Electron micrographs of longitudinal sections of renal tissue from rats experimentally intoxicated by paracetamol (PAR) and treated with coenzyme Q10 and N-acetylcysteine.** (A) CON-: negative control: intact tubules; (B) NAC + CoQ10 group that were treated associated with NAC and CoQ10: intact tubules and glomeruli; (C) CoQ10 group that were treated with coenzyme Q10: intact tubules and glomeruli; (D) NAC group that were treated with NAC: intact tubules and glomeruli. (PNG)

**S1 File. Raw data.** (XLSX)

## Acknowledgments

A sincere thanks to the technical staff of the Histotechnics and Innovation Laboratory (LHIn)/ IPTSP/UFG by their valuable contribution to the histological processing of the material, microtomy and histological slides, in particular Gisleine Fernanda França, laboratory technician.

## Author Contributions

**Conceptualization:** Rayanne Henrique Santana da Silva, Ana Flávia Machado Botelho.

**Formal analysis:** Rayanne Henrique Santana da Silva, Mariana de Moura, Larissa de Paula, Kelly Carolina Arantes, Marina da Silva, Jaqueline de Amorim, Marina Pacheco Miguel, Daniela de Melo e Silva, Marília Martins Melo, Ana Flávia Machado Botelho.

**Investigation:** Rayanne Henrique Santana da Silva, Marina Pacheco Miguel, Danieli Brolo Martins, Ana Flávia Machado Botelho.

**Methodology:** Rayanne Henrique Santana da Silva, Mariana de Moura, Larissa de Paula, Kelly Carolina Arantes, Marina da Silva, Jaqueline de Amorim, Marina Pacheco Miguel, Danieli Brolo Martins, Daniela de Melo e Silva, Marília Martins Melo, Ana Flávia Machado Botelho.

**Project administration:** Ana Flávia Machado Botelho.

**Supervision:** Ana Flávia Machado Botelho.

**Writing – original draft:** Rayanne Henrique Santana da Silva, Marina Pacheco Miguel, Danieli Brolo Martins, Daniela de Melo e Silva, Marília Martins Melo, Ana Flávia Machado Botelho.

**Writing – review & editing:** Rayanne Henrique Santana da Silva, Danieli Brolo Martins, Daniela de Melo e Silva, Marília Martins Melo, Ana Flávia Machado Botelho.

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
