## [Decision Letter · Decision Letter 0]

27 Apr 2023

PONE-D-23-03966Effects of coenzyme Q10 and N-acetylcysteine on experimental poisoning by paracetamol in Wistar ratsPLOS ONE

Dear Dr. BOTELHO,

Thank you for submitting your manuscript to PLOS ONE. After careful consideration, we feel that it has merit but does not fully meet PLOS ONE’s publication criteria as it currently stands. Therefore, we invite you to submit a revised version of the manuscript that addresses the points raised during the review process.

We look forward to receiving your revised manuscript.

Kind regards,

Jianhong Zhou

Staff Editor

PLOS ONE

Reviewers' comments:

Reviewer's Responses to Questions

**Comments to the Author**

1. Is the manuscript technically sound, and do the data support the conclusions?

Reviewer #1: Yes

Reviewer #2: Partly

Reviewer #3: Yes

2. Has the statistical analysis been performed appropriately and rigorously? 

Reviewer #1: Yes

Reviewer #2: Yes

Reviewer #3: Yes

3. Have the authors made all data underlying the findings in their manuscript fully available?

Reviewer #1: Yes

Reviewer #2: Yes

Reviewer #3: Yes

4. Is the manuscript presented in an intelligible fashion and written in standard English?

Reviewer #1: Yes

Reviewer #2: No

Reviewer #3: Yes

5. Review Comments to the Author

Reviewer #1: • The idea of the manuscript is important because it discusses the effect of one of the most usable medications for analgesics that we used daily to relieve the pain resulting from headaches or fever without the supervision of a physician, which leads to liver and kidney diseases later.

• The manuscript is well written, and the introduction section is focusing on the theme of the manuscript.

• The statistical analysis has been performed appropriately and rigorously.

• Please refer to the following comments in the reviewed attached manuscript file.

1- Include the page number for the manuscript.

2- Include the “line numbers” to allow reviewers to easily mention the comments.

3- In the results, the authors should consider the ordering in writing the text to be aligned with the figures and the graphs name, as in “Hematological and serum biochemistry evaluation “an increase in erythrocytes, hematocrit, and hemoglobin in the CON+ group and evidenced thrombocytopenia in the group of animals intoxicated with PAR and treated with NAC (Figure 1).”, while in the figure1 they wrote as follows: (A) Mean and standard deviation of hematocrit (HCT), (B) erythrocytes and (C) hemoglobin.

4- The authors should organize each section in the manuscript by adding numbers to the section and subtitles as follows:

1. Introduction.

2. Materials and Methods.

2.1. Experimental Design.

5- In the discussion section, the fourth paragraph “Although PAR overdose produces ROS and changes DNA, and lipid peroxidation [8,23]. in the present study, no differences were observed between the groups”. The highlighted dot should be corrected with coma.

6- Table 2: please can you rewrite it in the proper way.

Reviewer #2: - Why were 31 rats divided into five groups?

- It is only necessary to use the full spelling of IP once, followed by the abbreviation.

- Suggest to standardise the term use for PAR - Paracetamol or acetaminophen Or Tylenol?

- The author stated in the statistical analysis that the experimental design is completely randomised. How is the experimental design performed randomisation? Aren't the groups assigned to certain treatments?

RESULTS:

- In haematological and serum biochemistry evaluation results, how do ALT and AST levels found represent liver morphological status?

- Please correct the statement where it said 'it was possible'. The observation has been made, so the results should be definite, not 'possible'.

E.g.:

1. It was possible to observe a significant increase of creatinine in the NAC group compared to NAC+CoQ10 group (Figure 2).

2. The results obtained by the analysis demonstrate the mean ± SD (standard deviation) of the groups, where it was possible to observe that there was no statistical difference between them.

- The scale bar in fig. 4-5 is small. It is suggested that a clearer scale bar be provided.

- Figure 6: Add an arrow as an indicator to the results, such as intact mitochondria/normal nuclei/fat droplets, etc.

Other comments:

The study is descriptive rather than comprehensive. It would be ideal if some parameters, such as ROS levels, were added to confirm the observation (since PAR overdose did not show any differences between the treatment and control groups.); as well as antioxidant activities (glutathione levels etc).

A mechanism study would be preferable to corroborate the findings, particularly on the mechanism of hepatoprotection action of CoQ10 by measuring NRF2, proinflammatory cytokines or mitochondrial enzymes as postulated in the discussion. Several arguments are speculative, with no experimental evidence to back them up.

Reviewer #3: The study was well conducted by the authors;however,there are some concerns to revise that are described below

For Absrtct;

It is better to add one or two as introduction before the aim of study

Keywords ubiquinol ; ubiquinone

- to anyone that is not familiar with the subject these keywords are difficult - I don’t expect you to go deep in the chemistry of CoQ10

, but the reader should be able to understand all words

For Introduction;

The introduction section resumes the existing knowledge regarding antioxidant action of

NAC and CoQ10 .

However, as the importance of the topic, it is recommended to expand and update the literature.

For Method ;

The method section appears well organized with the relevant paper, but Comet assay was elaborated extensively and it is recommended to be reduced to some extent.

Minor comments ;

Discussion

In the fifth paragraph

No significant alteration were observed in GGT,FA and ALT

What does FA mean?

In the eighth paragraph

Nrf2 expression

Abbreviation should be explained to readers.

6. PLOS authors have the option to publish the peer review history of their article (what does this mean?). If published, this will include your full peer review and any attached files.

Reviewer #1: No

Reviewer #2: No

Reviewer #3: No

---

## [Author Response · Author response to Decision Letter 0]

2 Jun 2023

Authors are grateful for all suggestions and have addressed all reviewers’ comments:

Reviewer #1:

1- Include the page number for the manuscript 

We appreciate the suggestion and have included the number of pages in the article.

2- Include the “line numbers” to allow reviewers to easily mention the comments. 

Thanks for the suggestion and we included the lines in the article.

3- In the results, the authors should consider the ordering in writing the text to be aligned with the figures and the graphs name, as in “Hematological and serum biochemistry evaluation “an increase in erythrocytes, hematocrit, and hemoglobin in the CON+ group and evidenced thrombocytopenia in the group of animals intoxicated with PAR and treated with NAC (Figure 1).”, while in the figure1 they wrote as follows: (A) Mean and standard deviation of hematocrit (HCT), (B) erythrocytes and (C) hemoglobin.

We thank you for your comments and have already made the suggested changes.

Line 176: We changed the sentence “an increase in erythrocytes, hematocrit, and hemoglobin in the CON+ group and evidenced thrombocytopenia in the group of animals intoxicated with PAR and treated with NAC (Figure 1).” to “an increase in hematocrit, erythrocytes, and hemoglobin in the CON+ group and evidenced thrombocytopenia in the group of animals intoxicated with PAR and treated with NAC (Figure 1).” 

4- The authors should organize each section in the manuscript by adding numbers to the section and subtitles as follows:

1. Introduction.

2. Materials and Methods.

2.1. Experimental Design. 

Thanks for the suggestion and the sections of the manuscript have been organized by adding numbers and subheadings.

5- In the discussion section, the fourth paragraph “Although PAR overdose produces ROS and changes DNA, and lipid peroxidation [8,23]. in the present study, no differences were observed between the groups”. The highlighted dot should be corrected with coma 

We thank you for your comments and have already made the suggested changes.

Line 331: We changed the sentence “Although PAR overdose produces ROS and changes DNA, and lipid peroxidation [8,23]. in the present study, no differences were observed between the groups” to “Although PAR overdose produces ROS and changes DNA, and lipid peroxidation [8,23], in the present study, no differences were observed between the groups”.

6- Table 2: please can you rewrite it in the proper way.

Table was rewritten according to suggestions from reviewer

Reviewer #2: 

 Why were 31 rats divided into five groups?

We had planned to use 35 animals distributed into the five groups, however the university couldn’t provide the necessary number of animals at the time, considering the pandemic situation. We also had two deaths occur due to a gavage mistake, these animals did not participate in the study.

- It is only necessary to use the full spelling of IP once, followed by the abbreviation. 

Thanks for the suggestion. All incorrect uses have been reviewed.

- Suggest to standardize the term use for PAR - Paracetamol or acetaminophen Or Tylenol? 

Thanks for the suggestion. Sentence was reviewed accordingly.

- The author stated in the statistical analysis that the experimental design is completely randomized. How is the experimental design performed at randomisation? Aren't the groups assigned to certain treatments?

We had previously defined the treatments that we would perform in the project (CON-, CON+, NAC, CoQ10, NAC+CoQ10) and divided the animals into the identified boxes. The choice of animals in each box was performed at random. The experimental design used was completely randomized and the data obtained are presented as mean and standard deviation. The Kolmogorov-Smirnov test was performed to define normality. Those classified as parametric were evaluated by ANOVA and Tukey's post-test. In the case of non-parametric variables, the Kruskal-Wallis test and Dunn's post test of multiple comparisons were used. Spearman's correlation test was used to assess the relationship between serum biomarker levels and histological lesion scores. In all analyses, differences were considered significant when p<0.05. Data were evaluated using GraphPad Prism 7.

RESULTS:

- In haematological and serum biochemistry evaluation results, how do ALT and AST levels found represent liver morphological status? 

Hepatocyte damage can be verified by measuring ALT and AST in blood serum. ALT is a cytoplasmic enzyme that catalyzes the pyruvate-forming reaction of gluconeogenesis and is involved in amino acid metabolism. Its half-life in rats ranges from three to four hours, and normal values for this species comprise 35.1 ± 13.3 U/L. Increases in the levels of this liver enzyme reflect cellular damage to hepatocytes and may be linked to drug intoxication, including PAR. In PAR intoxication, there is an increase in the production of the NAPQI, which is positively correlated with the administered amount of the drug. The accumulation of this metabolite contributes to the formation of peroxynitrite with consequent peroxidation, destruction of proteins and damage to hepatocytes. In the present study, ALT did not differ between groups, although it showed a mean similar to AST. It is possible to observe from the scatter plot that some animals, possible outliers, significantly increase the mean and standard deviation of these groups. This finding is associated with and justified by the susceptibility of each individual to PAR intoxication, since the great variability of individual response has been described through genetic and hereditary factors. The observation of the ALT parameter also revealed a significant increase in the NAC group in relation to CON-, which suggests that its protective action was not effective enough to heal or mitigate liver damage. AST, in turn, is a mitochondrial and cytoplasmic enzyme, present in the deamination reaction of aspartate into oxaloacetate, and exhibits significant concentrations in the heart, liver, kidney and skeletal muscle. The normal values of this enzyme, for the studied species, comprise 42.9 ± 10.1 U/L and the elevation of this parameter may be related to damage to hepatocytes. Macroscopy showed congestion and evidence of a multifocal to diffuse lobular pattern in the liver of animals in the CON+ and NAC groups, compatible with what is described in the literature for PAR intoxication without further changes in other organs.

- Please correct the statement where it said 'it was possible'. The observation has been made, so the results should be definite, not 'possible'.

E.g.: 1. It was possible to observe a significant increase of creatinine in the NAC group compared to NAC+CoQ10 group (Figure 2). 

2. The results obtained by the analysis demonstrate the mean ± SD (standard deviation) of the groups, where it was possible to observe that there was no statistical difference between them.

Thanks for the suggestions. All have been reviewed accordingly.

- The scale bar in fig. 4-5 is small. It is suggested that a clearer scale bar be provided.

Thanks a lot for the suggestion. Scales have been modified for better visualization.

- Figure 6: Add an arrow as an indicator to the results, such as intact mitochondria/normal nuclei/fat droplets, etc.

Thanks a lot for the suggestion. Figure 6 has been revised.

Other comments:

The study is descriptive rather than comprehensive. It would be ideal if some parameters, such as ROS levels, were added to confirm the observation (since PAR overdose did not show any differences between the treatment and control groups.); as well as antioxidant activities (glutathione levels etc).

A mechanism study would be preferable to corroborate the findings, particularly on the mechanism of hepatoprotection action of CoQ10 by measuring NRF2, proinflammatory cytokines or mitochondrial enzymes as postulated in the discussion. Several arguments are speculative, with no experimental evidence to back them up.

We unfortunately don’t have the funds to further investigate the mechanisms. We are applying in our country for new funding in order to continue the research and provide such answers in the future. 

Reviewer #3: 

The study was well conducted by the authors; however, there are some concerns to revise that are described below

For Abstract;

It is better to add one or two as introduction before the aim of study

Thanks a lot for the suggestion. A paragraph has been added to the abstract. 

Line 21: “Paracetamol (PAR) is a drug widely used in human and veterinary medicine as an analgesic and antipyretic, often involved in cases of intoxication. The most common clinical signs result from damage to red blood cells and hepatocytes, and this intoxication is considered a model for the induction of acute liver failure”.

Keywords: ubiquinol ; ubiquinone

To anyone that is not familiar with the subject these keywords are difficult - I don’t expect you to go deep in the chemistry of CoQ10, but the reader should be able to understand all words

We are grateful for the suggestion. Keywords were revised.

Antioxidant; hepatotoxicity; hepatoprotective; oxidative damage; paracetamol; rats

For Introduction;

The introduction section resumes the existing knowledge regarding antioxidant action of NAC and CoQ10. However, as the importance of the topic, it is recommended to expand and update the literature.

Thanks for the suggestion. The introduction has been modified with updated articles.

For Method ;

The method section appears well organized with the relevant paper, but Comet assay was elaborated extensively and it is recommended to be reduced to some extent. Minor comments 

Thanks a lot for the suggestion. The text about comet assay has been shortened.

Discussion

In the fifth paragraph No significant alteration were observed in GGT, FA and ALT

What does FA mean? 

Thanks a lot for the suggestion. 

Line 123, 337: FA has been replaced by ALP

Line 363 was added Nrf2 means nuclear factor erythroid 2-related factor 2

After the first round of revisions, the following requirements were answered. Please let us know if they are sufficient.

Manuscript was revised accordingly.

We wish to acknowledge "Coordenação de Aperfeiçoamento de Pessoal de Nível Superior"—Finance Code 001" for the fellowship to Rayanne Henrique Santana da Silva. Also, we acknowledge the financial support for publication from FAPEG (Research Support Foundation of the State of Goiás), grant 06/2018.

The graphs for globulin, bilirubin and ALT were added to the supporting information and all “data not shown” were excluded.

All captions were inserted immediately after the first paragraph in which the figure is cited. Figure files were uploaded separately.

The graphs for globulin, bilirubin and ALT were added to the supporting information and all “data not shown” were excluded.

We don’t have a data repository subscription. But all data used can be made available as supportive material in Excel spreadsheets. 

Thank you for your consideration. We look forward to hearing from you.

Sincerely,

---

## [Decision Letter · Decision Letter 1]

21 Jul 2023

PONE-D-23-03966R1Effects of coenzyme Q10 and N-acetylcysteine on experimental poisoning by paracetamol in Wistar ratsPLOS ONE

Dear Dr. BOTELHO,

Thank you for submitting your manuscript to PLOS ONE. After careful consideration, we feel that it has merit but does not fully meet PLOS ONE’s publication criteria as it currently stands. Therefore, we invite you to submit a revised version of the manuscript that addresses the points raised during the review process.

Kind regards,

Yasmina Abd‐Elhakim

Academic Editor

PLOS ONE

Journal Requirements:

Reviewers' comments:

Reviewer's Responses to Questions

**Comments to the Author**

1. If the authors have adequately addressed your comments raised in a previous round of review and you feel that this manuscript is now acceptable for publication, you may indicate that here to bypass the “Comments to the Author” section, enter your conflict of interest statement in the “Confidential to Editor” section, and submit your "Accept" recommendation.

Reviewer #1: All comments have been addressed

Reviewer #2: All comments have been addressed

2. Is the manuscript technically sound, and do the data support the conclusions?

Reviewer #1: Yes

Reviewer #2: Yes

3. Has the statistical analysis been performed appropriately and rigorously? 

Reviewer #1: Yes

Reviewer #2: Yes

4. Have the authors made all data underlying the findings in their manuscript fully available?

Reviewer #1: Yes

Reviewer #2: Yes

5. Is the manuscript presented in an intelligible fashion and written in standard English?

Reviewer #1: Yes

Reviewer #2: No

6. Review Comments to the Author

Reviewer #1: The study was well conducted by the authors, and the authors have adequately addressed the comments raised in a previous round of review.

So I suggest that the manuscript is now acceptable for publication.

Reviewer #2: I believe that it is worth mentioning that 35 animals were distributed into the five groups, and later only 31 mice survived.

Liver Morphology is often examined through the analysis of the liver's histological features. The authors are looking at the levels of liver enzymes, which are indications of liver function. Please ensure that the terminology you're using in line197 is accurate.

7. PLOS authors have the option to publish the peer review history of their article (what does this mean?). If published, this will include your full peer review and any attached files.

Reviewer #1: No

Reviewer #2: No

---

## [Author Response · Author response to Decision Letter 1]

4 Aug 2023

Authors are grateful for all suggestions and have addressed all reviewers’ comments:

Reviewer #1:

1- The study was well conducted by the authors, and the authors have adequately addressed the comments raised in a previous round of review.

So I suggest that the manuscript is now acceptable for publication.

Thank you very much for your considerations.

Reviewer #2: 

1- I believe that it is worth mentioning that 35 animals were distributed into the five groups, and later only 31 mice survived.

Thank you for the suggestion. A sentence has been added to the abstract. 

Line 29: “Thirty-five adult Wistar rats (Rattus novergicus albinus) were randomly assigned to five groups, and thirty-one of these survived the treatments.”

Line 83: “Thirty-five adult male Wistar rats (Rattus novergicus albinus) were used, weighing on average 200-300g, distributed in five groups, and thirty-one of these survived the treatments. Animals were acquired at the Federal University of Goiás Bioterium.”

2-Liver Morphology is often examined through the analysis of the liver's histological features. The authors are looking at the levels of liver enzymes, which are indications of liver function. Please ensure that the terminology you're using in line197 is accurate.

Thanks for the suggestion, we changed the terminology presented in line 196.

Line 196: “To evaluate the liver function ALT and AST were measured.”

---

## [Editor Report · Decision Letter 2]

7 Aug 2023

Effects of coenzyme Q10 and N-acetylcysteine on experimental poisoning by paracetamol in Wistar rats

PONE-D-23-03966R2

Dear Dr. BOTELHO,

We’re pleased to inform you that your manuscript has been judged scientifically suitable for publication and will be formally accepted for publication once it meets all outstanding technical requirements.

Kind regards,

Yasmina Abd‐Elhakim

Academic Editor

PLOS ONE
---

## [Editor Report · Acceptance letter]

10 Aug 2023

PONE-D-23-03966R2 

Effects of coenzyme Q10 and N-acetylcysteine on experimental poisoning by paracetamol in Wistar rats 

Dear Dr. Botelho:

I'm pleased to inform you that your manuscript has been deemed suitable for publication in PLOS ONE. Congratulations! Your manuscript is now with our production department. 

Kind regards, 

on behalf of

Prof. Dr. Yasmina Abd‐Elhakim 

Academic Editor

PLOS ONE